# FewGAD: Few-Shot Enhanced Graph Anomaly Detection via Generative Contrastive Learning

## Abstract

Graph anomaly detection (GAD) is critical in domains such as fraud detection, cybersecurity, and social network monitoring. However, existing approaches face two major challenges: the inherent scarcity of labeled anomalies in practical scenarios, and the widespread reliance on graph augmentation, which often distorts anomaly semantics and undermines model robustness. To address these issues, we propose FewGAD, a framework that leverages limited anomaly labels to enhance contrastive discrimination through high-order subgraph sampling without augmentation. By avoiding augmentation-induced distortion, this design fundamentally improves the robustness and semantic validity of learned representations, thereby enabling clearer separation between normal and anomalous nodes. Furthermore, a kernel density estimation mechanism expands the utility of scarce labels, enhancing data efficiency and strengthening anomaly discrimination under few-shot settings. Extensive experiments on five benchmark datasets demonstrate that FewGAD consistently surpasses state-of-the-art unsupervised and few-shot GAD methods, achieving an average AUC gain of 6.2%.

## 1 Introduction

In recent years, graph neural networks (GNNs) have made breakthrough progress in graph learning tasks such as node classification, connection prediction, and recommendation systems (Scarselli et al., 2008; Wu et al., 2020; Kipf & Welling, 2016; Veličković et al., 2017). With its collaborative modeling ability of structural relationships and node attributes, it has gradually become the core method of graph data mining. One of the tasks that has received wide attention is Graph Anomaly Detection (GAD). Its goal is to identify abnormal individuals that significantly deviate from most nodes in attribute characteristics or connection patterns (Noble & Cook, 2003; Akoglu et al., 2015). Such nodes often correspond to key risk targets in multiple real-world scenarios, such as fake accounts with abnormal connection relationships or abnormal behavior in social networks (Jia et al., 2017; Li et al., 2022), fraudulent accounts with suspicious fund flow characteristics in financial transaction networks (Pourhabibi et al., 2020; Hilal et al., 2022; Motie & Raahemi, 2024), and user nodes with abnormal scoring behavior in e-commerce platforms (Ma et al., 2021; Gao et al., 2023).

Recent research in GAD has largely shifted toward unsupervised approaches, such as DOMINANT (Ding et al., 2019) and AnomalyDAE (Fan et al., 2020). These methods leverage graph autoencoders or reconstruction-based mechanisms to model the structural and attribute patterns of nodes, enabling the identification of potential anomalies without requiring labels. Among these approaches, contrastive learning (He et al., 2020; Zheng et al., 2022a; Li et al., 2023) has emerged as a prevailing paradigm in unsupervised graph anomaly detection, as datasets typically contain a substantial number of normal graph instances. By constructing positive and negative sample pairs, contrastive methods enable the model to learn discriminative representations that effectively distinguish between normal and anomalous nodes. This paradigm has demonstrated strong potential in improving anomaly detection performance and has inspired the development of numerous contrastive frameworks tailored for graph data (Liu et al., 2021; Duan et al., 2023; Lu et al., 2024).

It has been acknowledged by domain experts that acquiring a limited set of labeled anomalies is practical in real-world scenarios (Akoglu et al., 2015; Liu et al., 2024; Qiao et al., 2025). Such

Figure 1: Comparison of RWR and High-order Neighborhood Sampling. The left shows an RWR subgraph with weak semantic relevance and low contrastive signal. The right illustrates our high-order subgraph capturing semantically coherent neighbors. Guided by few-shot anomaly labels, our method enhances negative sampling, enlarges contrastive margins, and improves anomaly detection.

labels can be utilized as valuable prior knowledge to guide the training process, thereby holding great promise for improving the effectiveness of graph anomaly detection models (Zhang et al., 2022b; Ding et al., 2021; Satorras & Estrach, 2018; Chen et al., 2023). However, to develop a few-shot detection framework, we identify the following main challenges:

**1.** *Shallow Neighborhood Bias:* Existing methods (Ding et al., 2019; Zheng et al., 2021; Jin et al., 2021; Zhang et al., 2022a) commonly rely on random walk with restart (Tong et al., 2006) or graph augmentation–based sampling to extract subgraphs. These approaches often fail to capture the most discriminative structural patterns around anomalous nodes, especially when nodes exhibit sparse connectivity or irregular local structures. Additionally, augmentation-based sampling may inadvertently alter node features or local structures, resulting in semantic distortion. As illustrated in Fig. 1, many anomalies are located beyond the immediate (first-order) neighborhood, emphasizing the need for subgraph sampling strategies that cover higher-order structures and preserve semantic coherence.

**2.** *Weak Contrastive Boundaries*: Unsupervised contrastive objectives frequently yield weak decision boundaries, as positive and negative pairs in latent space are insufficiently separated. This issue arises because the absence of anomaly labels leads to negative samples that partially overlap with positives, limiting discriminability (Qiao et al., 2024; Zhou et al., 2024). In the few-shot setting, although labeled anomalies are scarce, we find they can serve as crucial anchors to guide contrastive learning. By leveraging these labels, we selectively construct negatives that are maximally dissimilar to positives, thereby enlarging the contrastive margin. This alleviates the intrinsic limitations of unsupervised contrastive learning and improves representation separability for anomaly detection.

To address these challenges, we propose FewGAD, a novel framework for few-shot anomaly detection. FewGAD leverages scarce anomaly labels to enhance contrastive discrimination while preserving semantic fidelity. To overcome the shallow neighborhood bias, we introduce a high-order neighborhood sampling module that constructs informative node-centric subgraph pairs without relying on graph augmentation, thereby avoiding anomaly distortion and capturing richer contextual structures. In addition, a kernel density estimation (KDE)-based module is employed under a local consistency constraint to expand the representation of labeled anomalies and mitigate the effects of data scarcity. This design strengthens the model's discriminative power, improves generalization to unseen anomaly types, and enables the selection of informative negative samples for contrastive learning. Extensive experiments on five benchmark datasets demonstrate that FewGAD consistently outperforms state-of-the-art unsupervised and few-shot graph anomaly detection methods. To summarize, the main contributions are as follows:

- We study the practical problem of few-shot graph anomaly detection, addressing the scarcity of labeled anomalies in real-world graphs.

- We propose FewGAD, a novel framework that leverages limited labels to enhance contrastive discrimination, incorporating high-order subgraph sampling to capture richer structural context without graph augmentation.

- We introduce a KDE-based module under local consistency to expand labeled information, generate discriminative negative samples, and improve generalization and robustness, with extensive experiments demonstrating superior performance over state-of-the-art methods.

## 2 RELATED WORK

**Contrastive learning-based GAD.** Contrastive learning (He et al., 2020; Zheng et al., 2022a; Li et al., 2023), a self-supervised approach that derives meaningful representations from unlabeled data, has gained considerable traction in graph anomaly detection due to its ability to reduce reliance on manual labeling and associated costs. CoLA (Liu et al., 2021) is the first method to introduce contrastive learning into this domain, capturing anomaly-aware representations by contrasting nodes with their local subgraph constructed through a random walk procedure. Building on this idea, ANEMONE (Jin et al., 2021) estimates node anomalous scores through the contrast of node & node and node & ego-net multi-scale instance pairs, for more comprehensive anomaly estimation. Another approach is presented by SAMCL (Hu et al., 2023), which detects anomalous nodes via the subgraph-aligned contrastive learning across multiple views of the graph to enhance detection robustness. Sub-CR (Zhang et al., 2022a) is a self-supervised framework that combines multi-view contrastive learning with attribute reconstruction to detect anomalies in attributed networks. It leverages local-global contrastive views to encode structural and attribute information, and uses a masked autoencoder to highlight nodes with high reconstruction errors as anomalies. Most recently, GRA-DATE (Duan et al., 2023) calculates the anomaly scores of nodes via contrastive learning among node-node,node-subgraph, and subgraph-subgraph multi-scale instance pairs between the original view and the augmentation view, enabling richer multi-scale anomaly characterization. However, all of the above methods operate in a fully unsupervised manner, often suffering from limited guidance and poor generalization.

**Few-shot Graph Learning.** The scarcity of labeled data in graph-based anomaly detection, due to costly manual annotation of rare events, is a major hurdle. Few-shot graph learning (Satorras & Estrach, 2018; Chen et al., 2023) and cross-network meta-learning (Ding et al., 2021; Long et al., 2024) are innovative paradigms that tackle this issue by leveraging minimal supervision for robust generalization. For instance, SemiGNN (Wang et al., 2019) is a semi-supervised graph neural network designed for fraud detection. It expands labeled data using social relations to generate additional unlabeled data. The model employs a hierarchical attention mechanism to capture correlations between different neighbors and views. GDN (Ding et al., 2021) introduces a meta-learning framework that uses a few labeled anomalies to capture transferable patterns across networks, enhancing the statistical separability between normal and abnormal nodes. In a similar vein, Meta-PN (Ding et al., 2022) adopts a meta-learning-driven label propagation strategy, which enables the generation of reliable pseudo-labels for unlabeled nodes and facilitates large receptive field learning during training. These approaches, however, typically depend on auxiliary domains or cross-network information, which may not always be available in practice. One such method is ANEMONE-FS (Zheng et al., 2022b), which constructs two multi-scale comparison networks to learn robust node-context relationships. By maximizing consistency for unlabeled nodes and minimizing it for labeled anomalies within each mini-batch.

## 3 METHODOLOGY

### 3.1 PRELIMINARIES

**Notations.** In this paper, we first define an attributed graph as $G = (V, E, \mathbf{A}, \mathbf{X})$, where $V$ is the node set, $E \subset V \times V$ is the edge set, $\mathbf{A} \in \mathbb{R}^{n \times n}$ denotes the adjacency matrix, and $\mathbf{X} \in \mathbb{R}^{n \times m}$ represents the $m$-dimensional node attributes, with $n = |V|$ denotes the number of nodes. **High-order Neighborhood Sampling.** Let's $\mathbf{S} = [\mathbf{s}^{(1)}, \mathbf{s}^{(2)}, .., \mathbf{s}^{(k)}] \in \mathbb{R}^{n \times k}$ denote the multi-hop structural influence matrix, where each $\mathbf{s}^{(k)} \in \mathbb{R}^n$ captures the structural signal strength of the $k$-hop neighborhood. Specifically, $\mathbf{s}_i^{(1)} = \sum_j A_{ij}$, and recursively $\mathbf{s}^{(k)} = \mathbf{A} \cdot \mathbf{s}^{(k-1)}$. For each node $v_i$, we identify its most structurally influential $k$-hop neighbor $\hat{v}_i = \arg\max_{j \in \mathcal{N}_{v_i}} \mathbf{s}_j^{(k)}$, where $\hat{v}_i$ serves as a pivotal component in forming the $k$-order subgraph.

**Problem Statement: Few-shot GAD.** Given an attributed graph $G = (V, E, \mathbf{A}, \mathbf{X})$ with nodes $v_1, \ldots, v_n$, we assume access to a few-shot set of labeled anomalous nodes $V_L \subset V$, where $|V_L| \ll |V|$. The objective is to learn an anomaly scoring function $f$ that leverages both the few-shot labeled anomalies and the abundant unlabeled nodes to assign each node $v_i$ an anomaly score $k_i = f(v_i)$.

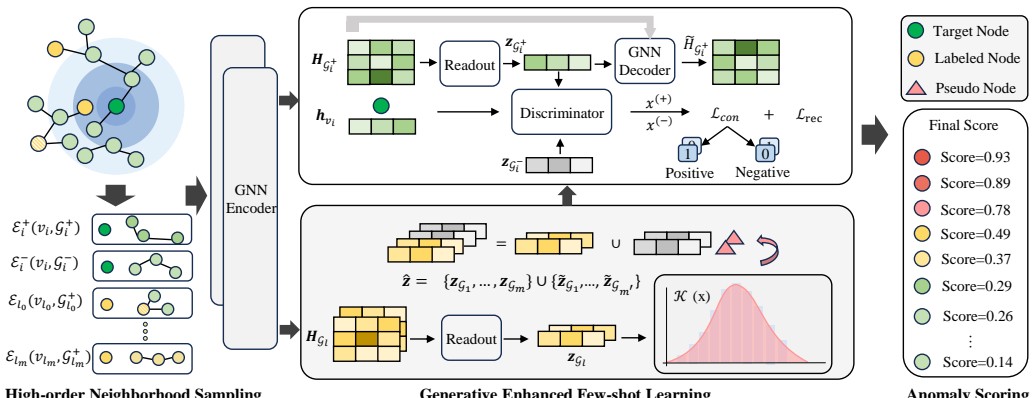

Figure 2: Overview of the FewGAD framework. The model consists of three components: (1) High-order Neighborhood Sampling constructs informative subgraphs by selecting high-order neighbors with strong influence, reducing structural bias. (2) Generative Enhanced Few-shot Learning leverages both labeled and unlabeled data to simulate diverse anomalies and improve generalization. (3) Anomaly Scoring computes the final anomaly score for each node based on the learned model.

The score $k_i$ quantifies the likelihood of $v_i$ being anomalous, and nodes are ranked accordingly, with those receiving higher scores identified as potential anomalies.

In few-shot settings, the limited connectivity of immediate neighborhoods often results in insufficient or noisy structural cues for anomaly detection. To mitigate this, we employ high-order neighborhood sampling to construct subgraphs that capture richer structural and semantic information. After embedding these subgraphs, the resulting positive and negative pairs exhibit a larger separation in the representation space, which effectively enlarges the contrastive margin. This enhanced discrimination enables contrastive learning to extract more expressive and robust anomaly representations, even under scarce supervision.

## 3.2 NODE-SUBGRAPH PAIRWISE CONTRAST

To enhance contrastive learning efficiency, we leverage the previously introduced high-order neighborhood sampling strategy to preprocess the graph and construct candidate contrastive subgraph pairs. Specifically, we extract the *most influential* and *least influential* $k$-order neighbor subgraphs from the graph structure. For each node $v_i$, we define a positive instance pair $\mathcal{E}_i^+ = (v_i, \mathcal{G}_i^+)$, where $\mathcal{G}_i^+$ is the $k$-order subgraph with the highest cumulative neighborhood influence connected to $v_i$. A corresponding negative pair $\mathcal{E}_i^- = (v_i, \mathcal{G}_j^-)$ is generated by randomly selecting another node $v_j$ ($j \neq i$) and using its least influential $k$-order subgraph $\mathcal{G}_j^-$.

Similarly, for labeled anomaly nodes $v_l \in \mathcal{V}_L$, we construct the labeled contrastive pair $\mathcal{E}_l = (v_l, \mathcal{G}_l^+)$, where $\mathcal{G}_l^+$ denotes the most influential $k$-order subgraph centered at $v_l$. All nodes and their corresponding subgraphs are embedded into a shared representation space during training to facilitate effective comparison and alignment between node-level and subgraph-level semantics.

**Subgraph Embedding.** To encode both the structure and node features within the high-order neighbor subgraphs, we adopt a Graph Convolutional Network (GCN) encoder (Kipf & Welling, 2016). For each subgraph $\mathcal{G}_i$ induced by the high-order neighborhood of node $v_i$, we denote its adjacency matrix and feature matrix as $\mathbf{A}_{\mathcal{G}_i}$ and $\mathbf{X}_{\mathcal{G}_i}$, respectively. The GCN propagation is defined as:

$$\mathbf{H}_{\mathcal{G}_i}^{(\ell)} = \sigma\left(\hat{\mathbf{D}}_{\mathcal{G}_i}^{-1/2}\hat{\mathbf{A}}_{\mathcal{G}_i}\hat{\mathbf{D}}_{\mathcal{G}_i}^{-1/2}\mathbf{H}_{\mathcal{G}_i}^{(\ell-1)}\mathbf{W}^{(\ell-1)}\right) \quad (1)$$

where $\hat{\mathbf{A}}_{\mathcal{G}_i} = \mathbf{A}_{\mathcal{G}_i} + \mathbf{I}$, and $\hat{\mathbf{D}}_{\mathcal{G}_i}$ is the degree matrix. The initial input is $\mathbf{H}_{\mathcal{G}_i}^{(0)} = \mathbf{X}_{\mathcal{G}_i}$. The activation function $\sigma(\cdot)$ (e.g., ReLU) is applied after each layer, and $\mathbf{W}^{(\ell-1)}$ is a trainable weight matrix.

After propagating through the GCN layers, we apply a readout function to obtain the subgraph-level representation. We use mean pooling over all node embeddings within $\mathcal{G}_i$: $\mathbf{z}_{\mathcal{G}_i} = \text{Readout}(\mathbf{H}_{\mathcal{G}_i}) = \frac{1}{|\mathcal{G}_i|}\sum_{j=1}^{|\mathcal{G}_i|}\mathbf{H}_{\mathcal{G}_i}(j)$, where $\mathbf{H}_{\mathcal{G}_i}(j)$ denotes the embedding of the $j$-th node in subgraph $\mathcal{G}_i$.

**Node Embedding.** For the node-level representation, we employ a multi-layer perceptron (MLP) to project node features into a latent embedding space. Unlike GCNs, this module does not incorporate structural information, enabling the model to focus purely on node attributes. The embedding process follows a layer-wise propagation scheme:

$$h_{v_i}^{(\ell)} = \sigma\left(h_{v_i}^{(\ell-1)}\mathbf{W}^{(\ell-1)}\right), \quad \ell = 1, \ldots, T \tag{2}$$

where $h_{v_i}^{(0)} = \mathbf{x}_{v_i}$ is the input feature of node $v_i$, $\mathbf{W}^{(l-1)}$ denotes the weight matrix at the $(l-1)$-th layer, and $\sigma(\cdot)$ is a nonlinear activation function (e.g., ReLU). The final node embedding $\mathbf{z}_{v_i} = h_{v_i}^{(T)}$ is obtained after $T$ layers of transformation.

**Generative Enhanced Few-shot Learning.** To improve the discriminative capability in the few-shot setting, we introduce a Generative-Enhanced Few-shot Learning mechanism. For each labeled anomaly node $v_l$, we encode its high-order subgraph with GCN and apply a readout to obtain subgraph embeddings $\mathbf{z}_{\mathcal{G}_l}$. Each embedding, denoted as $\mathbf{z}_{\mathcal{G}_l} \in \mathbb{R}^d$, represents a labeled anomaly node's subgraph and may be sparse or noisy. To enrich these representations, we apply KDE to estimate their distribution and sample synthetic embeddings accordingly.

$$\hat{f}(x) = \frac{1}{m\tau}\sum_{i=1}^{m}\mathcal{K}\left(\frac{x - \mathbf{z}_{\mathcal{G}_l}}{\tau}\right), \quad \mathcal{K}(x, x_i) = \exp\left(-2|x - x_i|_2^2\right) \tag{3}$$

We then sample $m'$ synthetic embeddings from the estimated density: $\tilde{\mathbf{z}}_1, \tilde{\mathbf{z}}_2, \ldots, \tilde{\mathbf{z}}_{m'} \sim \hat{f}(x)$. Finally, we form the augmented negative embedding set as: $\hat{\mathbf{z}} = \{\mathbf{z}_{\mathcal{G}_1}, \ldots, \mathbf{z}_{\mathcal{G}_m}\} \cup \{\tilde{\mathbf{z}}_{\mathcal{G}_1}, \ldots, \tilde{\mathbf{z}}_{\mathcal{G}_{m'}}\}$. where $\hat{h}$ denotes the combined set of original and synthetic anomaly subgraph embeddings.

**Loss of Contrastive Learning.** We adopt a bilinear function to measure the relation between node embeddings $\mathbf{z}_{v_i}$ and high-order neighbor-subgraph embeddings $\mathbf{z}_{\mathcal{G}_i}$. The similarity scores for positive and negative pairs are given by:

$$x_i^{(+)} = \text{Bilinear}(\mathbf{z}_{v_i}, \mathbf{z}_{\mathcal{G}i}), \quad x_i^{(-)} = \text{Bilinear}(\mathbf{z}_{v_i}, \mathbf{z}_{\mathcal{G}j}), \quad \hat{x}_i^{(-)} = \text{Bilinear}(\mathbf{z}_{v_i}, \hat{\mathbf{z}}_l), \quad i \neq j, \ \hat{\mathbf{z}}_l \in \hat{h}. \tag{4}$$

For each node $v_i$, we compute multiple negative scores $\hat{x}_i^{(-)}$ by evaluating the similarity between its embedding $\mathbf{z}_{v_i}$ and the augmented negative subgraph embeddings $\hat{\mathbf{z}}_l \in \hat{\mathbf{z}}$ using a bilinear function. Since some augmented subgraphs may still carry normal characteristics, we refine the final negative score by combining the hardest negative with a clear reference score. The final score is:

$$x_i^{(-)} = \alpha \min_{j \in \{1, \ldots, \tilde{m}\}} \tilde{x}_{ij}^{(-)} + (1 - \alpha) \cdot x_i^{(-)} \tag{5}$$

where $\alpha$ balances the hardest negative and reliable normal subgraph. The contrastive learning loss uses Binary Cross-Entropy with a Sigmoid layer, as shown in Equation (6):

$$\mathcal{L}_{con} = -\sum_{i=1}^{n_B}(y_i \log(\sigma(x_i)) + (1 - y_i)\log(1 - \sigma(x_i))) \tag{6}$$

where $n_B$ is the batch size, $\sigma()$ denotes the Sigmoid function. The label $y_i$ indicates whether a sample pair is positive or negative: $y_i = 1$ if $v_i \in \mathcal{E}_i^+$, and $y_i = 0$ if $v_i \in \mathcal{E}_i^-$.

To preserve the semantic consistency between input features and learned representations, we reconstruct node features via a one-layer GCN decoder and define a reconstruction loss based on Mean Squared Error:

$$\mathcal{L}_{\text{rec}} = \frac{1}{n}\sum_{i=1}^{n}\|\hat{\mathbf{x}}_i - \mathbf{x}_i\|_2^2 \tag{7}$$

where $\hat{\mathbf{x}}_i$ is the reconstructed feature of node $v_i$. To optimize training, we combine the few-shot contrastive and reconstruction learning modules, defining the total training loss function as follows:

$$\mathcal{L} = \beta\mathcal{L}_{con} + \lambda\mathcal{L}_{rec} \tag{8}$$

where the parameters $\beta$ and $\lambda$ control the relative importance of the two modules.

**Anomaly Scoring.** To quantify node-level abnormality, we employ both contrastive and reconstruction-based indicators. The contrastive signal is derived from the similarity gap between positive and negative pairs. Specifically, Let $x_i^{(+)}$ and $x_i^{(-)}$ denote the similarity scores of node $v_i$ with its positive and negative counterparts, respectively. Normal nodes typically yield $x_i^{(+)} \approx 1$ and $x_i^{(-)} \approx 0$, whereas anomalies deviate from this pattern. We thus define $\varpi_i^{con} = x_i^{(-)} - x_i^{(+)}$. In addition, we assess reconstruction deviation to capture irregularities in attribute recovery. Specifically, the reconstruction error is given by $\varpi_i^{rec} = \|\mathbf{x}_i - \hat{\mathbf{x}}_i\|_2^2$. We then integrate the two scores into a unified anomaly indicator $\varpi_i = \varpi_i^{con} + \lambda \cdot \varpi_i^{rec}$. To improve robustness against stochastic sampling and training noise, we conduct $R$ evaluation rounds and compute the final anomaly score as the sum of the mean and standard deviation of scores across rounds. Specially, let $\mu_i = \frac{1}{R} \sum_{k=1}^R \varpi_i^{(k)}$ and $\sigma_i = \sqrt{\frac{1}{R} \sum_{k=1}^R \left( \varpi_i^{(k)} - \mu_i \right)^2}$, then the final anomaly score is given by:

$$\text{Score}(v_i) = \mu_i + \sigma_i \tag{9}$$

### 3.3 THEORETICAL ANALYSIS

We analyze FewGAD from two key perspectives. A Subgraph Coverage Bound shows that our high-order sampling captures broader structures. A Controllability Analysis of embedding discrepancy explains how contrastive learning stabilizes representations in few-shot settings.

**Theorem 3.1 (Subgraph Coverage Bound for Max-Influence Sampling)** *Let $G = (V, E)$ be an undirected graph with $|V| = n$, adjacency matrix $\mathbf{A} \in \{0,1\}^{n \times n}$, and a set of $m \ll n$ anomalous nodes $S = \{v_1, \ldots, v_m\}$. Define the structural influence score of node $v_j$ as $\mathbf{s}_j^{(k)} = \sum_{i=1}^n [\mathbf{A}^k]_{ji}$, and its k-hop neighborhood as $\mathcal{N}^{(k)}(v_j) = \{v_l \mid [\mathbf{A}^k]_{jl} > 0\} \cup \{v_j\}$. For each $v_i \in S$, the proposed high-order sampling strategy selects $\hat{v}_i^{\mathrm{HI}} = \arg\max_{j \in \mathcal{N}^{(k)}(v_i)} \mathbf{s}_j^{(k)}$. Define the total coverage as:*

$$C_{\mathrm{HI}} = \left| \bigcup_{i=1}^m \mathcal{N}^{(k)} \left( \hat{v}_i^{\mathrm{HI}} \right) \right|$$

*where $\tilde{v}_i^{\mathrm{RW}} \in \mathcal{N}^{(k)}(v_i)$ is sampled uniformly at random. Then the following inequality holds:*

$$C_{\mathrm{HI}} \geq \max_{i=1,\ldots,m} \left| \mathcal{N}^{(k)} \left( \hat{v}_i^{\mathrm{HI}} \right) \right| - \alpha \binom{m}{2} \bar{N}^{(k)}$$

*where $\alpha \in (0,1)$ denotes the average pairwise neighborhood overlap, and $\bar{N}^{(k)} = \frac{1}{m} \sum_{i=1}^m \left| \mathcal{N}^{(k)} \left( \hat{v}_i^{\mathrm{HI}} \right) \right|$ is the mean size of the selected neighborhoods. Equality holds when the variance of $\mathbf{s}^{(k)}$ approaches zero, i.e., $\mathrm{Var}(\mathbf{s}^{(k)}) \to 0$, or $m \to n$, in which case max-influence selection becomes equivalent to random choice or full-graph coverage. See Appendix C.1 for the proof of the theorem.*

**Theorem 3.2 (Controllability of Cumulative Embedding Discrepancy)** *Let $f : \mathbb{R}^k \to \mathbb{R}$ be a Lipschitz continuous model with constant $L$, operating on subgraph embeddings $z \in \mathbb{R}^k$ derived from anomalous nodes in a graph $G = (V, E)$ with adjacency matrix $\mathbf{A}$. Given $m$ anomalous nodes $S = \{v_1, \ldots, v_m\}$, their subgraph embeddings $\{z_1, \ldots, z_m\}$ are obtained via high-order sampling. Let $\tilde{z}_j$, $j = 1, \ldots, m'$, be new embeddings generated via KDE with bandwidth $h$. The expected cumulative discrepancy satisfies:*

$$\mathbb{E}[\Delta = \sum_{j=1}^{m'} \min_{i=1,\ldots,m} |f(z_i) - f(\tilde{z}_j)|] \leq m' L \sqrt{C} h$$

*where $L$ is a Lipschitz continuity constant, which measures the rate of change of $f$'s output with respect to its input, reflecting the smoothness of the model, $C > 0$ is a constant related to the embedding dimension $k$, and the expectation is taken over the joint distribution of $\{\tilde{z}_j\}_{j=1}^{m'}$. This ensures that the total discrepancy between generated and original embeddings is controllably bounded, facilitating stable training in graph anomaly detection. See Appendix C.2 for the proof of the theorem.*

Table 1: Comparison of AUC-ROC and AUC-PR Results Across Unsupervised and Few-Shot Methods ( best in bold, second best underlined).

| Model | Cora | | Citeseer | | BlogCatalog | | Flickr | | ACM | |
|---|---|---|---|---|---|---|---|---|---|---|
| | AUC-ROC | AUC-PR | AUC-ROC | AUC-PR | AUC-ROC | AUC-PR | AUC-ROC | AUC-PR | AUC-ROC | AUC-PR |
| DOMINANT | 0.8124 | 0.3246 | 0.8267 | 0.3227 | 0.6465 | 0.0816 | 0.7454 | 0.1305 | 0.7986 | 0.1134 |
| AnomalyDAE | 0.8706 | 0.4373 | 0.8435 | 0.2765 | 0.7303 | 0.4348 | 0.7532 | 0.1548 | 0.8181 | 0.2573 |
| CoLA | 0.8942 | 0.4836 | 0.8786 | 0.4000 | 0.7800 | 0.2747 | 0.7468 | 0.2479 | 0.8322 | 0.3263 |
| ANEMONE | 0.8975 | 0.5223 | 0.9137 | 0.5148 | 0.6417 | 0.1056 | 0.6620 | 0.1242 | 0.8398 | 0.3409 |
| SL-GAD | 0.9030 | 0.5581 | 0.8135 | 0.3189 | 0.7691 | 0.4028 | 0.7803 | 0.4208 | 0.8186 | 0.2710 |
| GRADATE | 0.8421 | 0.4459 | 0.8231 | 0.2219 | 0.6058 | 0.1068 | 0.7181 | 0.1920 | 0.8231 | 0.2661 |
| AS-GAE | 0.6786 | 0.1288 | 0.6730 | 0.1482 | 0.4947 | 0.0545 | 0.5021 | 0.0554 | 0.5019 | 0.0336 |
| GDN | 0.7638 | 0.1763 | 0.7909 | 0.2470 | 0.5295 | 0.0651 | 0.5400 | 0.0673 | 0.7475 | 0.2353 |
| ANEMONE-FS | 0.9156 | 0.5206 | 0.9308 | 0.5238 | 0.7297 | 0.1780 | 0.7521 | 0.2550 | 0.8493 | **0.3608** |
| **ASD-HC-FS** | **0.9574** | **0.6093** | **0.9492** | **0.5639** | **0.8042** | **0.5430** | **0.8270** | **0.4219** | **0.8867** | 0.3552 |

## 4 EXPERIMENTS

In this section, we evaluate the effectiveness and efficiency of our proposed method by addressing the following questions: **Q1:** Can our method perform well in extreme few-shot or limited-label scenarios? **Q2:** How do different components contribute to overall performance? **Q3:** How sensitive and robust is our method to key hyperparameter changes?

### 4.1 EXPERIMENTAL SETTINGS

**Datasets.** We evaluate our proposed method and baselines on five widely used benchmark datasets, categorized into citation networks (Cora (McCallum et al., 2000), Citeseer (Lawrence et al., 1999), ACM (Sen et al., 2008)) and social networks (BlogCatalog, Flickr (Tang & Liu, 2009)). In citation networks, nodes represent documents and edges indicate citation links, with node features extracted from text content. In social networks, nodes denote users and edges represent relationships, with features derived from associated tags. We follow established strategies (Ding et al., 2019; Liu et al., 2021) to inject structural and attribute anomalies, as the datasets lack ground-truth labels. Details of the injection process and anomaly statistics are provided in App.D.1.

**Baselines.** We compare our proposed method with several representative baselines, including unsupervised approaches DOMINANT (Ding et al., 2019), AnomalyDAE (Fan et al., 2020), CoLA (Liu et al., 2021), ANEMONE (Jin et al., 2021), SL-GAD (Zheng et al., 2021), and GRADATE (Duan et al., 2023), as well as few-shot anomaly detection methods GDN (Ding et al., 2021) and ANEMONE-FS (Jin et al., 2021). These methods cover a wide range of modeling paradigms, such as graph autoencoders, contrastive learning, and generative strategies. Detailed descriptions of each baseline can be found in App. D.2.

**Implementations.** We adopt AUC-ROC and AUC-PR as evaluation metrics, where AUC-ROC evaluates overall discriminative capability, and AUC-PR is more indicative under class imbalance. For implementation, we employ a single-layer GCN as the encoder with a hidden size of 64, and train the model using the Adam optimizer (Kingma & Ba, 2014). The training schedule, learning rates, number of epochs, and neighborhood sampling parameters are adapted for each dataset to ensure stable performance. In all experiments, we set the number of labeled anomaly nodes to 10, and apply KDE-based sampling to augment anomaly representation. Further implementation details and hyperparameter settings are provided in the App. D.4.

### 4.2 PERFORMANCE ANALYSIS (RQ1)

In this section, we evaluate the performance of our proposed method for anomalous node detection in comparison with unsupervised methods and few-shot approaches. To ensure a consistent few-shot setting, all few-shot methods are provided with 10 labeled anomalous nodes as supervision.

**Advantage over Unsupervised GAD Methods.** We first compare our proposed FewGAD with unsupervised graph anomaly detection methods. As shown in Table 1, FewGAD consistently outperforms the baselines across multiple benchmark datasets, achieving superior results in both AUC-ROC and AUC-PR scores. For instance, on the Cora dataset, FewGAD achieves a 9.7% improvement over the best-performing unsupervised method, while on BlogCatalog and Flickr, the relative

Table 2: Ablation Study Results of FewGAD on Benchmark Datasets. We evaluate the contribution of three components: Contrastive learning (Con), Reconstruction objective (Rec), and Few-shot (Few).

Table 3: Comparison of AUC-ROC scores across varying few-shot label numbers $m$. Bold values indicate the highest results obtained under each setting.

| Con | Rec | Few | Cora | Citeseer | BlogCatalog | Flickr | ACM |
|---|---|---|---|---|---|---|---|
| ✓ | ✓ | ✓ | **0.9574** | **0.9492** | **0.8042** | **0.8270** | **0.8867** |
|  | ✓ | ✓ | 0.7778 *-18.8%* | 0.7710 *-18.8%* | 0.7435 *-7.5%* | 0.7439 *-10.0%* | 0.7472 *-15.7%* |
| ✓ |  | ✓ | 0.8687 *-9.3%* | 0.7998 *-15.7%* | 0.7575 *-5.8%* | 0.8077 *-2.3%* | 0.8361 *-5.7%* |
| ✓ | ✓ |  | 0.8936 *-6.7%* | 0.9013 *-5.0%* | 0.7661 *-4.7%* | 0.8147 *-1.5%* | 0.7896 *-10.9%* |

| Methods | Cora | Citeseer | BlogCatalog | Flickr | ACM |
|---|---|---|---|---|---|
| 1-shot | 0.9526 | 0.8257 | 0.7883 | 0.7342 | 0.8086 |
| 3-shot | 0.9441 | 0.9328 | 0.7949 | 0.8050 | 0.8337 |
| 5-shot | 0.9517 | 0.9220 | 0.7927 | 0.8111 | 0.8462 |
| 10-shot | **0.9574** | 0.9492 | 0.7990 | 0.8223 | 0.8867 |
| 15-shot | 0.9605 | **0.9503** | **0.8042** | **0.8349** | **0.8996** |

gains reach 11.3% and 8.5%, respectively. This performance gain demonstrates the effectiveness of integrating limited supervision into the contrastive framework, enabling FewGAD to better capture subtle anomalous patterns that are often overlooked by purely unsupervised approaches. The inferior performance of many unsupervised baselines can be attributed to several limitations. First, methods like DOMINANT and AnomalyDAE rely heavily on reconstruction-based objectives, which tend to underperform when anomalies are structurally similar to normal nodes or when graph sparsity is high. Additionally, although CoLA and GRADATE employ contrastive learning, their positive and negative sampling strategies are fixed or heuristic-based, lacking the adaptability needed to distinguish hard-to-detect anomalies. Furthermore, these methods are typically sensitive to the quality of graph structure or node attributes, making them less robust in real-world heterogeneous settings.

**Effectiveness in Few-Shot Anomaly Detection.** To further validate FewGAD's applicability in few-shot scenarios, we conduct experiments where only a few labeled anomalies are provided during training, as shown in Table 1. Compared with few-shot baselines like ANEMONE-FS and GDN, our method achieves notable improvements. This improvement can be attributed to the principled sampling of high-order substructures and the KDE-based negative instance generation strategy, which collectively enhance the model's generalization capacity under limited supervision. Few-GAD's advantage in few-shot settings stems from two key designs. First, its principled sampling of high-order subgraphs captures more informative and context-aware representations. Second, the KDE-based negative sampling adaptively generates harder contrastive pairs, enhancing learning under sparse supervision. In contrast, ANEMONE-FS relies on fixed sampling strategies, and GDN's meta-learning approach struggles with transferability in sparse or heterogeneous graphs.

## 4.3 ABLATION STUDY (RQ2)

To verify the effectiveness of each key component in FewGAD, we conduct an ablation study by introducing several model variants. Specifically, NoCon disables the contrastive learning module by setting $\beta = 0.0$, NoRec removes the reconstruction module by setting $\lambda = 0.0$, and NoFew excludes the few-shot labeled sample learning module. These variants allow us to systematically assess the contribution of each individual component to the overall performance, as shown in Table 2.

The ablation results demonstrate that all components of FewGAD contribute meaningfully to its overall effectiveness. Notably, removing the contrastive learning module (NoCon) leads to the most significant performance degradation, with an average AUC-ROC drop of 14.1%, including severe declines on Cora and Citeseer (18.8% and 18.8%, respectively), highlighting its critical role in enhancing the discriminative power of node representations. The reconstruction module (NoRec) also proves important, with an average drop of 6.6%, especially on Citeseer, indicating the value of structural reconstruction for anomaly identification. While the few-shot learning module (NoFew) results in a smaller average decrease of 3.7%, its contribution remains meaningful, validating its utility in effectively leveraging limited labeled anomalies to further improve detection performance.

## 4.4 SENSITIVITY ROBUSTNESS ANALYSIS (RQ3)

**Effect of the Size of Neighbor-Subgraph.** As shown in Fig. 3(a), with $k = 3$, increasing the subgraph size $t$ generally improves AUC, peaking around $t \in [15, 20]$ before declining. An exception

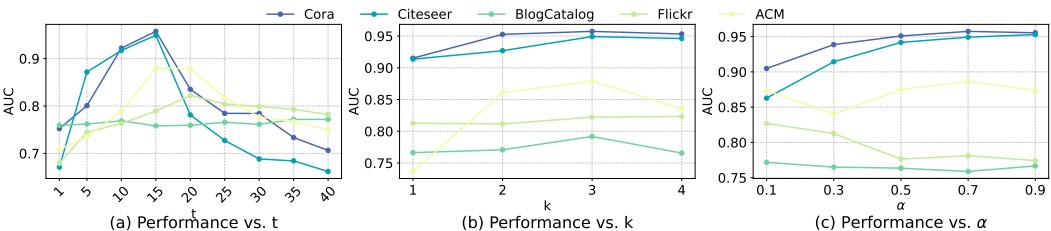

Figure 3: Parameter sensitivity of FewGAD on five benchmark datasets. (a) AUC performance under varying subgraph size $t$. (b) performance with different neighborhood orders $k$. And (c) examines the impact of the balance coefficient $\alpha$ on contrastive negative sampling.

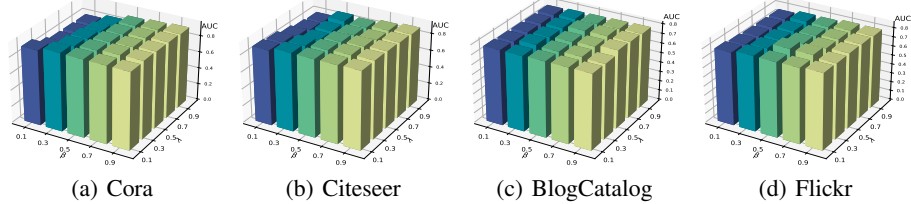

(a) Cora      (b) Citeseer      (c) BlogCatalog      (d) Flickr

Figure 4: AUC of ASD-HC-FS on Cora, BlogCatalog, and Flickr with varying $\beta$ (contrastive) and $\lambda$ (reconstruction) weights.

is BlogCatalog, where AUC continues to rise until $t > 35$, likely due to its high edge-node ratio and dense structure, making larger subgraphs more effective for capturing meaningful features.

**Effect of the High-Order of Neighbors.** Taking the Cora dataset as an example, while varying the neighborhood order $k$ from 1 to 4, the AUC value reaches its peak when $k = 3$. This trend is consistent across most datasets, where performance improves up to $k = 3$ but remains stable or slightly declines at $k = 4$, as shown in Fig. 3(b).

**Effect of the Balancing Factors.** We investigate the impact of balancing factors $\alpha, \beta, \lambda$. As shown in Fig. 4, increasing the contrastive loss weight $\beta$ improves AUC scores, with optimal performance when $\beta \in [0.3, 0.7]$. Conversely, a higher reconstruction loss weight $\lambda$ tends to degrade performance, with 0.1 being optimal. This highlights the importance of emphasizing contrastive signals over reconstruction for anomaly detection. Fig. 3(c) shows the effect of $\alpha$ under fixed $\beta$ and $\lambda$. On Cora, Citeseer, and ACM, increasing $\alpha$ enhances performance, with best results in $\alpha \in [0.7, 0.9]$. However, on noisier graphs like BlogCatalog and Flickr, performance is less sensitive or even declines, suggesting that strong auxiliary signals may hinder learning in complex networks.

**Effectiveness and Robustness under Few-shot Settings** We evaluate our model with varying numbers of labeled anomalies $m \in \{1, 3, 5, 10, 15\}$ across five datasets. As shown in Table 3, our method performs well even with very few labels, e.g., achieving 0.9526 AUC on Cora with only one labeled node. Performance steadily improves with more labels and tends to stabilize after $m = 10$, demonstrating both effectiveness and robustness. On complex graphs like BlogCatalog and Flickr, the model also shows consistent gains, confirming its adaptability across different structures.

## 4.5 CONCLUSIONS

In this paper, we propose FewGAD, a novel framework for few-shot graph anomaly detection. FewGAD integrates structural and attribute information through generative contrastive learning, enabling effective use of both labeled and unlabeled data. To mitigate local structural bias, we design a high-order neighborhood sampling module that constructs informative subgraph pairs, while a KDE-based module with local consistency enhances scarce anomaly representations and alleviates data sparsity. Experiments on five benchmark datasets show that FewGAD consistently surpasses state-of-the-art unsupervised and few-shot methods, demonstrating strong robustness and generalization across diverse graph scenarios.

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

## A    APPENDIX

## B    DERIVATION OF EQUATION 3.

The Gaussian kernel function is a widely used kernel in density estimation and machine learning due to its smoothness and locality properties. It measures the similarity between a point $x$ and a sample point $x_i$ based on their Euclidean distance. The general form of the Gaussian kernel is given by:

$$K(x, x_i) = \exp\left(-\frac{\|x - x_i\|_2^2}{2\tau^2}\right) \tag{10}$$

where $\tau$ is the bandwidth parameter. In our case, we fix the bandwidth as $h = 0.5$. Substituting this value into the Gaussian kernel formula, we obtain:

$$K(x, x_i) = \exp\left(-\frac{\|x - x_i\|_2^2}{2 \cdot (0.5)^2}\right) = \exp\left(-\frac{\|x - x_i\|_2^2}{0.5}\right) \tag{11}$$
$$= \exp\left(-2\|x - x_i\|_2^2\right)$$

## C    PROOFS

### C.1    THE PROOF OF THEOREM 1

**Lemma C.1 (Graph Heterogeneity)** *The graph $G$ is heterogeneous:* $Var(\mathbf{s}^{(k)}) = \frac{1}{n}\sum_{i=1}^{n}\left(\mathbf{s}_i^{(k)} - \bar{s}^{(k)}\right)^2 \geq \sigma^2$, *where* $\mathbf{s}_i^{(k)} = \sum_{j=1}^{n}[\mathbf{A}^k]_{ij}$, $\bar{s}^{(k)} = \frac{1}{n}\sum_i \mathbf{s}_i^{(k)}$, *and* $\sigma^2 > 0$.

**Proof C.1** *In real-world graphs (e.g., social networks), nodes have diverse roles, leading to varied $k$-hop influence $\mathbf{s}_i^{(k)}$. The variance $Var(\mathbf{s}^{(k)}) \geq \sigma^2$ holds as $\mathbf{s}_i^{(k)} = [\mathbf{A}^k \cdot \mathbf{1}]_i$ reflects structural differences, with $\sigma^2$ bounded away from zero for non-uniform graphs.*

$$Var(\mathbf{s}^{(k)}) = \frac{1}{n} \sum_{i=1}^{n} \left( s_i^k - \bar{s}^{(k)} \right)^2 \geq \sigma^2.$$

**Lemma C.2 (Anomalous Node Characteristics)** *Anomalous nodes have low influence: $\mathbf{s}_i^{(k)} \leq \tau < \bar{s}^{(k)}$, $\forall v_i \in S$, where $S$ is the set of anomalous nodes.*

**Proof C.2** *Anomalies (e.g., fraudsters) are often peripheral in graphs, with fewer $k$-hop connections. Thus, $\mathbf{s}_i^{(k)} = \sum_{j=1}^{n} [\mathbf{A}^k]_{ij} \leq \tau < \bar{s}^{(k)}$, as their paths to other nodes are limited compared to the average.*

$$\mathbf{s}_i^{(k)} = \sum_{j=1}^{n} [\mathbf{A}^k]_{ij} \leq \tau, \quad \tau < \frac{1}{n} \sum_{l=1}^{n} \mathbf{s}_l^{(k)}.$$

**Lemma C.3 (Bounded Neighborhood Size)** *The $k$-hop neighborhood size is bounded: $|N^{(k)}(v_i)| \leq \Delta_k \leq n$, $\forall v_i \in V$.*

**Proof C.3** *The $k$-hop neighborhood $N^{(k)}(v_i) = \{v_j \mid [\mathbf{A}^k]_{ij} > 0\} \cup \{v_i\}$ contains nodes reachable in $k$-hops. Since $G$ has $n$ nodes, $|N^{(k)}(v_i)| \leq \Delta_k \leq n$, where $\Delta_k$ depends on graph connectivity.*

$$|N^{(k)}(v_i)| = |\{j \mid [\mathbf{A}^k]_{ij} > 0\} \cup \{i\}| \leq \Delta_k \leq n.$$

**Lemma C.4 (Few-Shot Sparsity)** *The number of anomalous nodes is sparse: $m \leq \sqrt{n}$.*

**Proof C.4** *In few-shot anomaly detection, labeled anomalies are scarce relative to graph size. Thus, $m \ll n$, and $m \leq \sqrt{n}$ ensures sparsity, limiting overlap in sampled subgraphs.*

$$m \leq \sqrt{n}.$$

*Proof.* For each anomalous node $v_i \in S$, high-influence sampling identifies the node with maximum $k$-hop influence in its $k$-hop neighborhood:

$$\hat{v}_i^{\mathrm{HI}} = \arg \max_{j \in N^{(k)}(v_i)} \mathbf{s}_j^{(k)}, \quad \mathbf{s}_j^{(k)} = \sum_{l=1}^{n} [\mathbf{A}^k]_{jl}.$$

By Lemma C.2, $\mathbf{s}_i^{(k)} \leq \tau < \bar{s}^{(k)}$, indicating low influence for anomalous nodes. However, Lemma C.1 guarantees that the $k$-hop neighborhood $N^{(k)}(v_i)$ contains nodes with diverse influences due to graph heterogeneity ($Var(\mathbf{s}^{(k)}) \geq \sigma^2 > 0$). Consequently, the selected $\hat{v}_i^{\mathrm{HI}}$ satisfies:

$$\mathbf{s}_{\hat{v}_i^{\mathrm{HI}}}^{(k)} \geq \bar{s}^{(k)}.$$

Since the $k$-hop influence $\mathbf{s}_j^{(k)}$ correlates with the neighborhood size $|N^{(k)}(v_j)|$, it follows that:

$$|N^{(k)}(\hat{v}_i^{\mathrm{HI}})| \geq \bar{N}^{(k)}.$$

The term $\max_{i=1,\ldots,m} |N^{(k)}(\hat{v}_i^{\mathrm{HI}})|$ denotes the largest $k$-hop neighborhood among the high-influence neighbors of anomalous nodes. By Lemma C.3, this size is bounded:

$$\max_{i=1,\ldots,m} \left| N^{(k)}(\hat{v}_i^{\mathrm{HI}}) \right| \leq \Delta_k \leq n.$$

Nevertheless, Lemma C.1 implies that heterogeneity amplifies the neighborhood sizes of high-influence nodes, ensuring that the maximum is significantly larger than the average $\bar{N}^{(k)}$.

The adjustment term $\alpha \binom{m}{2} \bar{N}^{(k)}$ represents the expected connectivity contribution from all pairs of the $m$ anomalous nodes under a null model, where:

- $\bar{N}^{(k)} = \frac{1}{n}\sum_{i=1}^{n}|N^{(k)}(v_i)|$ is the average $k$-hop neighborhood size,

- $\binom{m}{2} = \frac{m(m-1)}{2}$ counts all pairs among $m$ nodes,

- $\alpha \in (0,1)$ is a significance parameter controlling the threshold for non-anomalous connectivity.

By Lemma C.4, the number of anomalous nodes is sparse ($m \leq \sqrt{n}$), which limits the total expected connectivity and validates the use of $\alpha\binom{m}{2}\bar{N}^{(k)}$ as a reasonable adjustment for typical graph behavior.

The Higher Criticism statistic $C_{\text{HI}}$ is designed to detect anomalous subgraphs by identifying extreme deviations in neighborhood sizes. Specifically, $C_{\text{HI}}$ evaluates the significance of the largest observed neighborhood sizes relative to their expected values under a null hypothesis. For each $v_i \in S$, the neighborhood size $|N^{(k)}(\hat{v}_i^{\text{HI}})|$ is a test statistic, and $C_{\text{HI}}$ emphasizes the most extreme deviation:

$$C_{\text{HI}} \geq \max_{i=1,\ldots,m}\left(\left|N^{(k)}(\hat{v}_i^{\text{HI}})\right| - \mathbb{E}[|N^{(k)}|]\right),$$

where $\mathbb{E}[|N^{(k)}|]$ is the expected neighborhood size under the null model. In the context of pairwise interactions among $m$ nodes, we approximate:

$$\mathbb{E}[|N^{(k)}|] \approx \alpha\binom{m}{2}\bar{N}^{(k)},$$

since $\binom{m}{2}\bar{N}^{(k)}$ estimates the total expected connectivity across all pairs, and $\alpha$ adjusts for the significance level. Thus:

$$C_{\text{HI}} \geq \max_{i=1,\ldots,m}\left|N^{(k)}(\hat{v}_i^{\text{HI}})\right| - \alpha\binom{m}{2}\bar{N}^{(k)}.$$

By Lemma C.1, graph heterogeneity ensures that some high-influence neighbors $\hat{v}_i^{\text{HI}}$ have large neighborhood sizes. By Lemma C.4, the sparsity of anomalous nodes ($m \leq \sqrt{n}$) supports the appropriateness of the adjustment term. Therefore, $C_{\text{HI}}$, as a statistic capturing the most significant neighborhood deviation, satisfies the inequality:

$$C_{\text{HI}} \geq \max_{i=1,\ldots,m}\left|N^{(k)}(\hat{v}_i^{\text{HI}})\right| - \alpha\binom{m}{2}\bar{N}^{(k)}.$$

## C.2 THE PROOF OF THEOREM 2

**Assumption C.1 (Lipschitz Continuity)** *The model $f : \mathbb{R}^k \to \mathbb{R}$ is Lipschitz continuous with constant $L$, i.e., for all $z, \tilde{z} \in \mathbb{R}^k$:*

$$|f(z) - f(\tilde{z})| \leq L\|z - \tilde{z}\|_2.$$

**Assumption C.2 (Bounded Embeddings)** *The subgraph embeddings $z_i$ and generated embeddings $\tilde{z}_j$ lie in a bounded subset of $\mathbb{R}^k$, i.e., there exists $B > 0$ such that $\|z_i\|_2, \|\tilde{z}_j\|_2 \leq B$.*

*Proof.* We establish the bound on the expected cumulative discrepancy $\mathbb{E}[\Delta]$ in several steps, leveraging the Lipschitz continuity of $f$, the KDE-based generation process, and the high-order sampling mechanism.

**Lemma C.5 (Lipschitz Bound on Individual Discrepancy)** *For any generated embedding $\tilde{z}_j$, the discrepancy $\Delta(\tilde{z}_j) = \min_{i=1,\ldots,m}|f(z_i) - f(\tilde{z}_j)|$ is bounded by:*

$$\Delta(\tilde{z}_j) \leq L\min_{i=1,\ldots,m}\|z_i - \tilde{z}_j\|_2.$$

**Proof C.5** *By Assumption C.1, the model $f$ satisfies:*

$$|f(z_i) - f(\tilde{z}_j)| \leq L\|z_i - \tilde{z}_j\|_2,$$

*for all $i = 1,\ldots,m$. Taking the minimum over all original embeddings:*

$$\Delta(\tilde{z}_j) = \min_{i=1,\ldots,m}|f(z_i) - f(\tilde{z}_j)| \leq \min_{i=1,\ldots,m}L\|z_i - \tilde{z}_j\|_2 = L\min_{i=1,\ldots,m}\|z_i - \tilde{z}_j\|_2.$$

**Lemma C.6 (KDE Perturbation Distance)** *For a generated embedding $\tilde{z}_j = z_{i_j} + h\epsilon_j$, where $i_j \sim Uniform\{1, \ldots, m\}$ and $\epsilon_j \sim \mathcal{N}(0, I_k)$, the minimum distance to the original embeddings satisfies:*

$$\min_{i=1,\ldots,m} \|z_i - \tilde{z}_j\|_2 \leq h\|\epsilon_j\|_2.$$

**Proof C.6** *Per Equation 3, the generated embedding is:*

$$\tilde{z}_j = z_{i_j} + h\epsilon_j,$$

*where $z_{i_j}$ is one of the original embeddings $\{z_1, \ldots, z_m\}$. The distance to any original embedding $z_i$ is:*

$$\|z_i - \tilde{z}_j\|_2 = \|z_i - (z_{i_j} + h\epsilon_j)\|_2.$$

*Since $z_{i_j} \in \{z_1, \ldots, z_m\}$, we evaluate the distance to $z_{i_j}$:*

$$\|z_{i_j} - \tilde{z}_j\|_2 = \|z_{i_j} - (z_{i_j} + h\epsilon_j)\|_2 = h\|\epsilon_j\|_2.$$

*Thus, the minimum distance is:*

$$\min_{i=1,\ldots,m} \|z_i - \tilde{z}_j\|_2 \leq \|z_{i_j} - \tilde{z}_j\|_2 = h\|\epsilon_j\|_2.$$

**Lemma C.7 (Expected Minimum Distance)** *The expected minimum distance between a generated embedding $\tilde{z}_j$ and the original embeddings satisfies:*

$$\mathbb{E}\left[\min_{i=1,\ldots,m} \|z_i - \tilde{z}_j\|_2\right] \leq h\sqrt{k}.$$

**Proof C.7** *From Lemma C.6, we have:*

$$\min_{i=1,\ldots,m} \|z_i - \tilde{z}_j\|_2 \leq h\|\epsilon_j\|_2.$$

*Taking expectations over the distribution of $\tilde{z}_j$, which includes the randomness of $i_j$ and $\epsilon_j$:*

$$\mathbb{E}\left[\min_{i=1,\ldots,m} \|z_i - \tilde{z}_j\|_2\right] \leq \mathbb{E}[h\|\epsilon_j\|_2] = h\mathbb{E}[\|\epsilon_j\|_2].$$

*Since $\epsilon_j \sim \mathcal{N}(0, I_k)$, the squared norm $\|\epsilon_j\|^2 = \sum_{l=1}^{k} \epsilon_{j,l}^2$ follows a chi-squared distribution $\chi^2(k)$:*

$$\mathbb{E}[\|\epsilon_j\|^2] = k, \quad Var(\|\epsilon_j\|^2) = 2k.$$

*By Jensen's inequality:*

$$\mathbb{E}[\|\epsilon_j\|_2] = \mathbb{E}[\sqrt{\|\epsilon_j\|^2}] \leq \sqrt{\mathbb{E}[\|\epsilon_j\|^2]} = \sqrt{k}.$$

*For large $k$, $\mathbb{E}[\|\epsilon_j\|_2] \approx \sqrt{k}$. Thus:*

$$\mathbb{E}\left[\min_{i=1,\ldots,m} \|z_i - \tilde{z}_j\|_2\right] \leq h\sqrt{k}.$$

**Lemma C.8 (Cumulative Discrepancy Aggregation)** *The expected cumulative discrepancy satisfies:*

$$\mathbb{E}[\Delta] \leq m'Lh\sqrt{k}.$$

**Proof C.8** *The cumulative discrepancy is:*

$$\Delta = \sum_{j=1}^{m'} \min_{i=1,\ldots,m} |f(z_i) - f(\tilde{z}_j)| = \sum_{j=1}^{m'} \Delta(\tilde{z}_j).$$

*Taking expectations:*

$$\mathbb{E}[\Delta] = \mathbb{E}\left[\sum_{j=1}^{m'} \Delta(\tilde{z}_j)\right] = \sum_{j=1}^{m'} \mathbb{E}[\Delta(\tilde{z}_j)].$$

*Since the $\tilde{z}_j$ are independently and identically distributed, we focus on $\mathbb{E}[\Delta(\tilde{z}_j)]$. By Lemma C.5:*

$$\Delta(\tilde{z}_j) \leq L \min_{i=1,\dots,m} \|z_i - \tilde{z}_j\|_2.$$

*Taking expectations:*

$$\mathbb{E}[\Delta(\tilde{z}_j)] \leq L\mathbb{E}\left[\min_{i=1,\dots,m} \|z_i - \tilde{z}_j\|_2\right].$$

*By Lemma C.7:*

$$\mathbb{E}\left[\min_{i=1,\dots,m} \|z_i - \tilde{z}_j\|_2\right] \leq h\sqrt{k}.$$

*Thus:*

$$\mathbb{E}[\Delta(\tilde{z}_j)] \leq Lh\sqrt{k}.$$

*Summing over $m'$ generated embeddings:*

$$\mathbb{E}[\Delta] = \sum_{j=1}^{m'} \mathbb{E}[\Delta(\tilde{z}_j)] \leq \sum_{j=1}^{m'} Lh\sqrt{k} = m'Lh\sqrt{k}.$$

With these lemmas established, we complete the proof of the theorem. From Lemma C.8, we have:

$$\mathbb{E}[\Delta] \leq m'Lh\sqrt{k}.$$

Set $\sqrt{C} = \sqrt{k}$, where $C = k$ is the dimension-related constant:

$$\mathbb{E}[\Delta] \leq m'L\sqrt{C}h.$$

# D  IMPLEMENTATION DETAILS

## D.1  DETAILED DATASET DESCRIPTION

Table 4: The statistics of the datasets.

| Dataset | Nodes | Edges | Attributes | Anomalies | Ratio |
|---|---|---|---|---|---|
| **Cora** | 2,708 | 5,429 | 1,433 | 150 | 5.5% |
| **Citeseer** | 3,327 | 4,723 | 3,703 | 150 | 4.5% |
| **BlogCatalog** | 5,196 | 171,743 | 8,189 | 300 | 5.8% |
| **Flickr** | 7,575 | 239,738 | 12,407 | 450 | 5.9% |
| **ACM** | 16,484 | 71,980 | 8,337 | 600 | 3.6% |

To evaluate the detection ability of our algorithm, anomaly ground truth in datasets is essential. However, since all experimental datasets lack ground-truth anomaly labels, we follow the anomaly injection strategies adopted in Ding et al. (2019); Liu et al. (2021), including structural anomaly injection and attribute anomaly injection. We randomly select $q$ nodes and induce $q$ connected subgraphs using a random walk approach and then transform them into $q$ fully connected subgraphs. Similarly, we randomly sample $q$ connected subgraphs with the same number of nodes as Set $T$, and select $k$ nodes as Set $C$. The attribute of each node in $T$ is perturbed based on the Euclidean distance between it and the randomly selected node in $C$. The number of anomalies in each dataset can be found in Table 4. A detailed introduction of these datasets is given as follows:

- Cora: The Cora dataset is a citation network where each node represents a scientific publication and edges denote citation relationships. Each paper is assigned a topic label and described by a bag-of-words feature vector.

- Citeseer: Citeseer is another citation network similar to Cora, where nodes represent research papers and edges indicate citations. Each paper is associated with a single label from a set of scientific categories and described using word-frequency features.

- BlogCatalog: BlogCatalog is a social network dataset where nodes correspond to bloggers and edges indicate social connections between them. Each user may have multiple associated interest labels, making this a multi-label classification scenario.
- Flickr: The Flickr dataset is collected from a photo-sharing social media platform, where nodes represent users and edges denote their social connections. Users are annotated with multiple interest categories based on the tags of the images they share.
- ACM: The ACM dataset is derived from the DBLP bibliographic database, forming a heterogeneous graph with nodes representing papers, authors, and research fields. Paper nodes are characterized by keyword-based features and are assigned to academic subject categories.

### D.2 MORE ABOUT THE BASELINES

- **DOMINANT** Ding et al. (2019) employs a deep graph autoencoder method and utilizes graph structure and features for detecting anomalous nodes in a graph.
- **AnomalyDAE** Fan et al. (2020) AnomalyDAE detects anomalies by measuring reconstruction errors through the complex interaction of network structure and node properties with dual autoencoders.
- **CoLA** Liu et al. (2021) is an anomaly detection algorithm targeting nodes, using GNN-based contrastive learning at node-subgraph level. It computes anomaly scores by evaluating representations from nodes and subgraphs in positive and negative instance pairs.
- **ANEMONE** Jin et al. (2021) is an anomalous node detection method based on Graph Neural Networks (GNN), aiming to identify graph anomalies using multi-scale patch and context-level contrastive learning. **ANEMONE-FS** builds on ANEMONE and extends it to few-shot scenarios with limited labeled anomalies.
- **SL-GAD** Zheng et al. (2021) is a self-supervised method that incorporates two components: generative attribute regression and multi-view contrastive learning. Generative attribute regression detect nodes that behave differently in the attribute space from the neighbors. In contrast, multi-view contrastive learning highlights the structural differences between a node and its neighbors.
- **GRADATE** Duan et al. (2023) is an anomalous node detection approach based on node-node, node-subgraph, and subgraph-subgraph multi-view contrastive learning.
- **GDN** Ding et al. (2021) is a GNN-based model for few-shot anomaly detection that identifies anomalous nodes, edges, or subgraphs using limited labeled data. It introduces deviation loss for training, and leverages cross-network meta-learning to enhance detection across various domains.

### D.3 COMPLEXITY ANALYSIS

**Time Complexity Analysis.** We analyze the computational complexity of each component in our framework as follows: We first compute the multi-hop structural influence matrix $\mathbf{S} = [\mathbf{s}^{(1)}, \ldots, \mathbf{s}^{(k)}]$, where each $\mathbf{s}^{(k)}$ is obtained recursively via $\mathbf{s}^{(k)} = \mathbf{A} \cdot \mathbf{s}^{(k-1)}$. This step requires $\mathcal{O}(k\eta N)$ time, where $\eta$ denotes the average node degree and $N$ is the total number of nodes.

Following this, we perform greedy high-order path sampling for each node based on the structural influence vector. This involves a local neighborhood search of path length $l$ and has time complexity $\mathcal{O}(N\eta(k + t))$, where $t$ denotes the final subgraph size used for representation learning.

For the contrastive learning module, the main cost lies in computing the similarity between positive and negative pairs in each batch. For a subgraph of size $t$, the per-node complexity is $\mathcal{O}(t^2)$, and thus for all training nodes, the overall complexity is $\mathcal{O}(Nt^2)$. During the inference phase, we perform $R$ evaluation rounds, the total time complexity becomes $\mathcal{O}(RNt^2)$.

### D.4 EXPERIMENTAL CONFIGURATION AND HYPERPARAMETER TUNING

We conduct all experiments on a powerful GPU setup featuring an RTX 4090 (24GB), which provides the necessary computational resources for handling graph data.

**Implementations.** We employed a single-layer GCN as the encoder, with the hidden dimension fixed at 64. We adopt the Adam optimizer Kingma & Ba (2014) to streamline and optimize the model's training process. During training, the batch size is set to 200. The model is trained for 100 epochs on Cora, Citeseer, and Books, and for 200 epochs on BlogCatalog, Fickr, and ACM. The learning rate is 0.001 for Cora, Citeseer, Flickr, and Books, 0.003 for BlogCatalog, and 0.0005 for ACM. The evaluation phase consisted of 256 rounds. Additionally, the size of the $k$-order neighbors and the configuration of the neighbor subgraphs very depending on the dataset used. For Cora, Citeseer and ACM, the parameter $t$ is set to 15; for Flickr, it is set to 20; for BlogCatalog, it is set to 35. We keep the value of $k$ fixed at 3rd order. For each dataset, the number of labeled anomaly nodes is set to 10, and the number of samples generated by KDE is 100 for all datasets except for Blog, which generates 15 samples. Table lists all the hyperparameters used in our model along with their corresponding search spaces. During training, we conduct a grid search to identify the model configuration that achieves the highest AUROC score on the validation set.

Table 5: FewGAD Hyperparameter Tuning Ranges

| Hyperparameter | Distribution |
|---|---|
| learning rate | $5e^{-4}$-$1e^{-1}$ |
| epochs | 100-200 |
| the order of neighbors | [1,2,3,4] |
| high-order neighbor subgraph size | 10-40 |
| KDE sample size | 10-100 |
| $\alpha$ | [0.1, 0.3, 0.5, 0.7, 0.9] |
| $\beta$ | [0.1, 0.3, 0.5, 0.7, 0.9] |
| $\gamma$ | [0.1, 0.3, 0.5, 0.7, 0.9] |

### D.5 MORE ABOUT RESULTS

To further validate the robustness of our method, we report Precision-Recall Area Under Curve (AUC-PR) scores across all benchmark datasets in Table 6. Compared with both unsupervised and few-shot baselines, FewGAD consistently achieves superior performance, with the highest AUC-PR scores in most cases. This demonstrates the method's ability to maintain high precision and recall, particularly under imbalanced anomaly detection settings where AUC-PR is a more informative metric than AUC-ROC.

Table 6: Comparison of AUC-PR Results Across Unsupervised and Few-Shot Methods (best in bold, second best underlined).

| Model | Cora | Citeseer | BlogCatalog | Flickr | ACM |
|---|---|---|---|---|---|
| DOMINANT | 0.3246 | 0.3227 | 0.0816 | 0.1305 | 0.1134 |
| AnomalyDAE | 0.4373 | 0.2765 | 0.4348 | 0.1548 | 0.2573 |
| CoLA | 0.4836 | 0.4000 | 0.2747 | 0.2479 | 0.3263 |
| ANEMONE | 0.5223 | 0.5148 | 0.1056 | 0.1242 | 0.3409 |
| SL-GAD | 0.5581 | 0.3189 | 0.4028 | 0.4208 | 0.2710 |
| GRADATE | 0.4459 | 0.2219 | 0.1068 | 0.1920 | 0.2661 |
| GDN | 0.1763 | 0.2470 | 0.0651 | 0.0673 | 0.2353 |
| ANEMONE-FS | 0.5206 | 0.5238 | 0.1780 | 0.2550 | **0.3608** |
| **ASD-HC-FS** | **0.6093** | **0.5639** | **0.5430** | **0.4219** | 0.3552 |

From the box plots (5)comparing anomaly scores of normal and anomalous nodes, we observe that FewGAD not only achieves clearer separation between the two classes but also yields a more concentrated score distribution for normal nodes, as indicated by the shorter box length. This reduced variance in normal node scores suggests that FewGAD provides more stable and consistent anomaly scoring, reducing false positives and enhancing overall detection reliability compared to CoLA, which shows greater score variability among normal nodes.

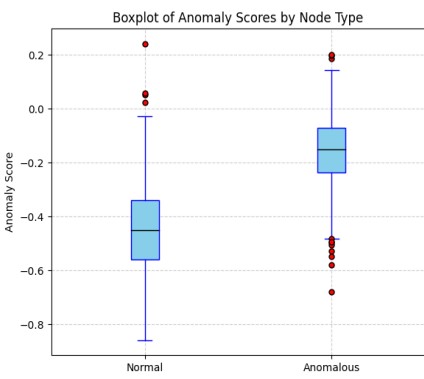

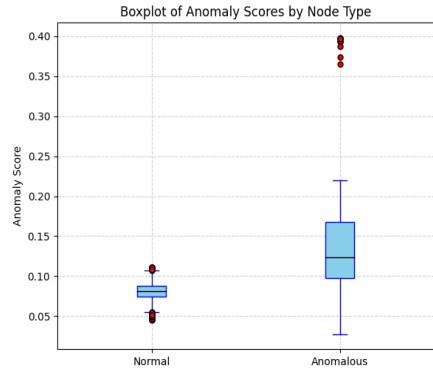

(a) Score Distribution of Normal and Anomalous Nodes by CoLA

(b) Score Distribution of Normal and Anomalous Nodes by FewGAD

Figure 5: AUC-ROC performance under different settings.

# E SOCIETAL IMPACT

This paper proposes a novel framework for graph anomaly detection under the few-shot setting. Our goal is to effectively detect anomalies with only a limited number of labeled abnormal samples, making the approach well-suited for scenarios with scarce annotated data. This enhances anomaly detection capabilities across various graph-structured domains such as social networks, industrial control systems, and transportation networks. Regarding ethical considerations, we do not currently anticipate any significant ethical concerns or potential for adverse societal impacts.

# F LIMITATION

While FewGAD achieves strong performance and efficient inference, it has two notable limitations. First, the training phase involves multi-hop structural influence computation and KDE-based negative sampling, which introduce additional time and memory costs compared to simpler sampling strategies. Although manageable on moderate-scale graphs, this overhead may become a bottleneck on large or dynamic graphs. Second, the model relies on a small number of labeled anomalies to guide the contrastive view construction. While the few-shot assumption is realistic in many real-world applications, the method's effectiveness may degrade if the labeled samples are scarce, noisy, or poorly distributed in the anomaly space.

