# OpenReview forum: "FewGAD: Few-Shot Enhanced Graph Anomaly Detection via Generative Contrastive Learning"
_ICLR.cc/2026/Conference — ICLR 2026 Conference Withdrawn Submission_

### Official Review · Reviewer_p4RX · 2025-10-25

**Soundness:** 3
**Presentation:** 3
**Contribution:** 2
**Rating:** 2
**Confidence:** 4

**Summary:**

This paper introduces FewGAD, a framework for few-shot graph anomaly detection that aims to overcome challenges in anomaly label scarcity and pitfalls of graph augmentation-based contrastive learning. The method replaces augmentation with high-order neighborhood sampling to generate discriminative subgraph pairs and leverages kernel density estimation (KDE) to enhance anomaly representation with limited labeled data. The authors provide theoretical analysis, implementation details, and extensive empirical evidence on five benchmark datasets, reporting improvements over strong unsupervised and few-shot GAD baselines in both AUC-ROC and AUC-PR.

**Strengths:**

1. The paper targets the realistic and critical setting where only a handful of anomaly labels are available, an under-explored but highly relevant regime for anomaly detection in graphs.

2. FewGAD forgoes potentially distortion-prone graph augmentations in favor of high-order neighborhood sampling, shown schematically in Figure 1, to build more semantically consistent subgraph pairs for contrastive learning. This addresses a well-known problem with existing augmentation-based methods and is a thoughtfully motivated contribution.

3. The use of KDE to synthesize harder negatives from scarce labeled anomalies efficiently expands the negative pool and could help prevent representation collapse or degenerate boundaries.

4. The anomaly scoring boxplots (Figures 5(a)/5(b)) demonstrate clearer separation and lower score variance for normal nodes under FewGAD relative to CoLA, providing compelling evidence for the framework’s discriminative improvement.

**Weaknesses:**

1. All benchmark datasets are label-free and require synthetic anomaly injection. The precise design of these injections could strongly affect model rankings. There is insufficient analysis of how assumptions made in the injection process, e.g., perturbation methods, structure/attribute ratios, influence FewGAD’s and competitors’ performance. This limits claims about “real-world” transfer, as models may specialize to, or overfit, the specific anomaly generation strategy.

2. Despite extensive ablations, the analysis is almost exclusively positive. Key questions go unaddressed: How fragile are the gains when labeled anomalies are noisy or mislocalized in feature space? What is the impact when anomalies are clustered vs. dispersed? How would FewGAD fare if (a) the labeled anomalies are not representative, i.e., drawn from an outlier subcluster, or (b) if negative augmentation by KDE produces points lying too close to normal representations? A more adversarial, stress-tested evaluation is needed to build trust.

3. The KDE-based embedding generation is described, but the precise size, distribution, and “hardness” selection of the negatives are only briefly mentioned in Section 3.2. The selection of $\alpha$ is described as a hyperparameter, yet there is little systematic exploration of how negative hardness or overlap with positives changes qualitative performance. Likewise, the linkage between theoretical bound and empirical negative sampling robustness is loose; the bound in Theorem 3.2 is population-level and may not fully capture practical collapse when the KDE-synthesized negatives are not well-placed.

4. Although a limitations section is present, the discussion is cursory and omits potential negative effects such as: propagation of anomaly detection errors, e.g., bias toward labeling minority substructures as anomalous), and increased resource usage (which may disadvantage graph learning in constrained deployments. For instance, the approach increases computational cost (see D.3) but does not address mitigation strategies beyond “manageable on moderate-scale graphs.”

5. The mathematical definitions, especially surrounding high-order neighborhood influence, notation for subgraphs, and the contrastive loss, are sometimes dense or inconsistently notated (Section 3.1, 3.2). For example, the indices for positive and negative subgraphs are overburdened with superscripts and hats, occasionally making equations hard to parse. The connection between the theoretical constructs (Theorem 3.1) and the concrete algorithm as implemented is not always clearly delineated.

**Questions:**

1. Can the authors systematically investigate the model’s robustness under different subgraph sizes and KDE sample counts, especially for graphs with highly skewed degree distributions? Do major performance drops occur if the subgraph size or KDE bandwidth is mis-tuned?

2. Given the synthetic nature of anomalies, how do FewGAD and baselines perform under alternative anomaly injection strategies (e.g., community-based anomalies, attribute shuffling, random edge insertions)? Could FewGAD overfit to artifacts of the injection pipeline rather than real anomaly structure?

3. Can you provide analysis or concrete examples where FewGAD underperforms, such as when anomaly features are weak, anomalies are tightly clustered, or labeled anomalies are not representative of the anomaly distribution?

4. How does the model perform on calibration-sensitive metrics, such as false positive rate at low recall or top-k precision on very imbalanced datasets? Would FewGAD be stable for deployment in real-world, time-critical scenarios?

---

### Official Review · Reviewer_3mwT · 2025-10-26

**Soundness:** 2
**Presentation:** 2
**Contribution:** 2
**Rating:** 4
**Confidence:** 4

**Summary:**

The paper introduces FewGAD, a few-shot graph anomaly detection framework designed to work with limited labeled anomalies. Specifically, it employs high-order subgraph sampling and kernel density estimation to construct positive-negative pairs and address the label scarcity problem.

**Strengths:**

1. This paper proposes a high-order subgraph sampling method to enhance contrastive learning.

2. A kernel density estimation mechanism is proposed to expand the utility of scarce labels.

2. The paper is easy to follow and achieves better performance than the used baselines.

**Weaknesses:**

1. In the introduction section, the motivation is not clearly stated.

2. The authors should discuss the difference between the proposed method and the recent semi-supervised GAD methods.

3. To demonstrate the effectiveness of the proposed high-order sampling, an ablation study is needed.

4. There are some grammatical errors, and the authors should carefully review the entire manuscript, such as the last sentence in the related work.

**Questions:**

Please see the weaknesses.

---

### Official Review · Reviewer_FW5x · 2025-10-27

**Soundness:** 2
**Presentation:** 2
**Contribution:** 2
**Rating:** 2
**Confidence:** 4

**Summary:**

This paper proposes a method for few-shot graph anomaly detection. Compared to unsupervised graph anomaly detection, a few labelled anomalous nodes are provided during training. To utilize these labelled nodes, the authors propose to employ contrastive learning based on high-order neighborhood sampling and KDE-based generation.

**Strengths:**

1. The studied problem, which is graph anomaly detection, is important and challenging.
2. The proposed high-order neighborhood sampling capture richer structural and semantic information.
3. To overcome the scarcity of labeled anomalies, a kernel density estimation mechanism is proposed.

**Weaknesses:**

1. The proposed few-shot setting where only labelled anomalies are available is not practical. When getting labelled anomalies in practice, it will also get lots of normal nodes as normal nodes dominates the whole graph.

2. The novelty of this paper is limited as it marginally improves the current contrastive learning for graph anomaly detection.

3. In the related work section, other types of graph anomaly detection methods should also be discussed to provide a comprehensive review.

4. Are there any specific reasons to use bilinear function instead of InfoNCE for contrastive learning? It has not been justified anywhere in the paper.

5. More datasets should be employed for comparison, especially these containing real anomalies.

**Questions:**

Please see weaknesses

---

### Official Review · Reviewer_oEJ9 · 2025-10-30

**Soundness:** 2
**Presentation:** 3
**Contribution:** 2
**Rating:** 4
**Confidence:** 5

**Summary:**

This paper focuses on the few-shot graph anomaly detection setting and proposes FewGAD, which introduces a high-order neighborhood sampling technique to enrich negative pairs using the limited labeled anomalies for contrastive learning. Experiments on five benchmark datasets with synthetic anomalies demonstrate the effectiveness of FewGAD.

**Strengths:**

(1)The paper proposes a novel high-order subgraph extraction method for contrastive learning and applies it to the few-shot GAD setting. The proposed approach is well-motivated and clearly presented.
(2)The authors provide theoretical insights to support the proposed sampling mechanism.

**Weaknesses:**

(1) This paper lacks the dedicated few-shot methods evaluations for GAD, some supervised methods, such as BWGNN [1]  ,GHRN  [2] or other reprsentive supervised methods, can be adapted to the few-shot setting by treating all remaining training nodes as normal and training the model using these nodes along with a few labeled abnormal nodes given the overwhelming presence of normal nodes.  Therefore, it is important to compare them with FewGAD to highlight its advantages. A recently published few-shot GAD method with publicly available code should also be included for comparison to ensure a more comprehensive and fair evaluation.
[1] Rethinking graph neural networks for anomaly detection. ICLR, 2022.
[2] Addressing heterophily in graph anomaly detection: A perspective of graph spectrum. WebConf, 2023.
[3] MetaGAD: Meta Representation Adaptation for Few-Shot Graph Anomaly Detection. DSAA, 2024.

(2) The experiments with varying numbers of few-shot abnormal nodes for the strong or representative competing methods should be included to evaluate the robustness of the proposed approach to ensure a fair and comprehensive evaluation. Evaluating the method solely by varying the number of few-shot samples does not provide a compelling demonstration of its effectiveness.

(3) The proposed high-order neighborhood sampling introduces considerable computational overhead. Please include a runtime comparison between the proposed method and representative baseline methods to support its efficiency. Additionally, its scalability to large-scale graphs may be limited and should be further discussed.

(4) As shown in Fig. 5 in the Appendix, FewGAD does not demonstrate a notable improvement in enlarging the margin between normal and abnormal samples. Additional evaluations on more datasets are needed to substantiate the effectiveness.

**Questions:**

See above **Weaknesses**

---

### Note · Authors · 2025-12-04

I have read and agree with the venue's withdrawal policy on behalf of myself and my co-authors.